# Analgesic Efficacy of Non-Steroidal Anti-Inflammatory Drug Therapy in Horses with Abdominal Pain: A Systematic Review

**DOI:** 10.3390/ani13223447

**Published:** 2023-11-08

**Authors:** Gerardo Citarella, Vanessa Heitzmann, Elisabeth Ranninger, Regula Bettschart-Wolfensberger

**Affiliations:** Section of Anaesthesiology, Department of Clinical Diagnostics and Services, Vetsuisse Faculty, University of Zurich, 8057 Zurich, Switzerland; vheitzmann@vetclinics.uzh.ch (V.H.); elisabeth.ranninger@uzh.ch (E.R.); rbettschart@vetclinics.uzh.ch (R.B.-W.)

**Keywords:** NSAIDs, equine, analgesia, colic

## Abstract

**Simple Summary:**

The use of non-steroidal anti-inflammatory drugs, that primarily act via the inhibition of COX isomers, is one of the most common therapeutic means to control abdominal pain in horses. However, these drugs can elicit gastrointestinal side effects. Drugs that are more selective for COX-2 inhibition are considered to cause less adverse effects. Despite some physiological effects of the COX-2 isoform, it is mainly induced by inflammatory processes, whereas the COX-1 isoform has a protective role and is considered to be constitutive. Despite the availability of several non-steroidal anti-inflammatory drugs with varying degrees of COX isoform inhibition, non-selective molecules remain the most frequently prescribed non-steroidal anti-inflammatory drugs. This is likely because the analgesic effect achieved by COX-2-selective drugs is not considered sufficient. To date, the scientific evidence concerning the analgesic efficacy of non-steroidal anti-inflammatory drugs in the treatment of abdominal pain in horses still remains uncertain. This systematic review showed that the current scientific literature cannot adequately justify the therapeutic choice of one non-steroidal anti-inflammatory drug over another for the treatment of abdominal pain in horses. Therefore, prospective randomised blinded clinical trials are deemed necessary to elucidate the analgesic efficacy of non-steroidal anti-inflammatory drugs in the treatment of abdominal pain in horses.

**Abstract:**

This systematic review aimed to identify the evidence concerning the analgesic efficacy of non-steroidal anti-inflammatory drugs to treat abdominal pain in horses, and to establish whether one non-steroidal anti-inflammatory drug could provide better analgesia compared to others. This systematic review was conducted following the “Systematic Review Protocol for Animal Intervention Studies”. Research published between 1985 and the end of May 2023 was searched, using three databases, namely, PubMed, Embase, and Scopus, using the words equine OR horse AND colic OR abdominal pain AND non-steroidal anti-inflammatory drug AND meloxicam OR flunixin meglumine OR phenylbutazone OR firocoxib OR ketoprofen. Risk of bias was assessed with the SYRCLE risk of bias tool, and level of evidence scored according to the Oxford Centre for Evidence-based Medicine. A total of 10 studies met the inclusion criteria. From those only one study judged pain with a validated pain score, and a high risk of bias was identified due to the presence of selection, performance, and “other” types of bias. Therefore, caution is required in the interpretation of results from individual studies. To date, the evidence on analgesic efficacy to determine whether one drug is more potent than another regarding the treatment of abdominal pain in horses is sparse.

## 1. Introduction

The use of non-steroid anti-inflammatory drugs (NSAIDs) is an indispensable aid in the treatment of visceral disorders in horses suffering from colic or post-castration abdominal pain. Over time, new molecules have been developed to limit NSAID-related side effects while preserving their analgesic and anti-inflammatory effects. NSAIDs have a different degree of inhibition of COX-isoforms depending on the type of molecule [1]. Recently, research has focused more on molecules with a high selectivity towards the inhibition of COX-2 (inducible isoform), which is responsible for the trigger of pain and inflammation in response to injury [2]. As such, the functionality of COX-1 (constitutive isoform), which is responsible for maintaining protective and reparative physiological mechanisms, is preserved [3].

In horse medicine, colic, defined as acute paroxysmal abdominal pain, represents one of the most frequent conditions encountered in clinical practice [4]. The treatment, which can evolve either as medical or surgical therapy, will most likely involve the use of NSAIDs [5,6]. A recent study, examining the proportion of NSAID prescriptions in equine practice, found that the most frequently prescribed NSAIDs for the treatment of colic in the UK, USA, and Canada were flunixin meglumine and phenylbutazone (traditional NSAIDs) [7]. Furthermore, earlier studies have confirmed similar findings in South Africa [8].

It is interesting to note that the prescribing trend is still formally linked to the use of traditional molecules (non-selective COX isoform) despite the availability of newer molecules such as meloxicam and firocoxib, which are designed to more specifically target the COX-2 isoform to avoid undesirable gastrointestinal side effects [9]. Apparently, this derives from a certain degree of scepticism towards the analgesic potency of some newer-generation NSAIDs, particularly firocoxib (entirely COX-2-selective) [9].

Considering the high rate of prescriptions of NSAIDs and their role in the analgesia management of abdominal pain in horses, it is crucial to elucidate the analgesic efficacy of the different classes of NSAIDs in this species. Clinical practice should be guided by evidence-based research. Systematic reviews build a connection between medical research and health care practice, and answer clinically relevant questions based on the evidence of all relevant literature regarding a specific research question.

For this reason, the aim of this systematic review is (i) to identify, synthetise, and evaluate the evidence concerning the analgesic efficacy of the NSAIDs available to treat abdominal pain in horses, and (ii) to establish, if possible, whether there is a NSAID that could provide better analgesia compared to others.

## 2. Materials and Methods

This systematic review was conducted following the “Systematic Review Protocol for Animal Intervention Studies (SYRCLE)” [10].

### 2.1. Disease/Health Problem and Population/Species

Studies were included if they investigated the effect of NSAIDs in adult horses (aged >6 months) with naturally occurring or experimentally induced colic and post-castration abdominal pain.

### 2.2. Interventions/Exposure

For this review, the administration of at least one of the following NSAIDs was considered as inclusion intervention: meloxicam and/or flunixin meglumine and/or phenylbutazone and/or firocoxib and/or ketoprofen, with specified dosage and route of administration in horses suffering from experimentally induced or naturally occurring colic- or castration-related abdominal pain. The administration had to be in a controlled (versus placebo) manner or in a comparative manner (one NSAID of interest compared to another one).

### 2.3. Control Population

For ethical reasons, most modern clinical trials investigating analgesic drugs do not include a control group. Therefore, the current study also included studies with no control group.

However, experimental studies conducted with a control group (absence of the selected intervention) were considered in this systematic review.

### 2.4. Outcome Measures

Pain score after administration of NSAIDs (mandatory)Clinical parameters such as heart rate (HR) and respiratory rate (RR) (if present).

### 2.5. Search Method

The investigation was conducted on three databases according to SYRCLE guidelines.

MEDLINE via PubMed;Embase;Scopus.

The search strategy was conducted according to the step-by-step search guide [11], and consisted in this string:

(equine OR horse) AND (colic OR abdominal pain) AND non-steroidal anti-inflammatory drug AND (meloxicam OR flunixin meglumine OR phenylbutazone OR firocoxib OR ketoprofen).

This string was adapted according to the search rules/code of the database used. All publications from 1985 to the end of May 2023 were searched.

### 2.6. Selection of Studies

Two reviewers independently screened the results of the search output. The first selection phase consisted of the evaluation of the titles and abstracts of the studies. Then, the second phase consisted of a careful reading of the full text of the selected papers.

The selected papers were analysed for their strength of evidence according to the Oxford Centre for Evidence-based Medicine [12]. The scoring to assess the quality of evidence consisted of 3 quality levels: the highest quality level (I-LoE) was awarded to papers including evidence from systematic reviews; quality level II (II-LoE) to papers with evidence obtained from properly designed, randomised controlled trials; and quality level III (III-LoE) to papers with evidence coming from non-randomised trials or experimental studies. Discrepancies between reviewers were resolved by including the judgement of a third person, and conclusions were drawn following a critical discussion between the reviewers.

### 2.7. Inclusion and Exclusion Criteria

Inclusion criteria:

We included controlled studies on either experimental or client-owned adult horses (aged > 6 months) that compared the analgesic efficacy of two or more NSAIDs, or one or more NSAIDs and a control group in horses with acute colic/abdominal pain. Only publications in English with at least abstract and title available were included. Only papers scoring I-LoE, II-LoE, and III-LoE for the quality of evidence were included in this systematic review.

Exclusion criteria:

We excluded studies with models of chronic abdominal pain (>6 weeks). Studies with ponies, miniature horses, and donkeys were excluded due to the potential for differing pathophysiological responses to colic. Studies concerning the effect of paracetamol or metamizole were also excluded due to the nature of those drugs identified as non-classical NSAIDs.

### 2.8. Data Extraction and Management

Details of the eligible studies were independently extracted by the two reviewers.

Data extracted were the following:Authors, title, year of publication, and journal;Number of horses in intervention and control groups (if present);NSAIDs used, disease natural or experimentally induced;Outcome measures (pain scoring);Excluded animals.

### 2.9. Assessment of Risk of Bias in the Included Studies

The two reviewers independently scored the selected studies regarding the risk of bias using a modified SYRCLE’s risk of bias tool [13]. Discrepancies were resolved by asking an additional person and with a critical discussion between the reviewers.

The modified tool to assess the included studies consisted of 10 signalling questions defined to analyse 6 types of bias: selection bias, performance bias, detection bias, attrition bias, reporting bias, and other biases. The review authors’ judgements about each risk of bias item for each included study and for each included variable question with a “yes” indicated a low risk of bias; “no” indicated a high risk of bias; and “unclear” indicated an unclear risk of bias. Questions number 4 and 9 were adapted to the need of this systematic review while question number 6 was judged as not applicable. A detailed description can be found in Table 1.

## 3. Results

The total amount of papers found was 22 (PubMed), 147 (Scopus), and 116 (Embase). After removing the 41 duplicates, 244 papers were screened as eligible. The first selection considering title and abstract included 18 papers. Of these, 10 were excluded. After full text examination (Figure 1), another two studies that met the inclusion criteria were found from references cited, not detected via the initial string. Finally, a total of 10 studies were included in this systematic review.

### 3.1. Characteristics of the Included Studies

A total of 10 studies were judged eligible for this systematic review; 4 of them referred to castration-related abdominal pain and the rest were either experimental or clinical trials regarding colic-related abdominal pain. Detailed information is provided in Table 2.

Of the castration studies, one consisted in the comparison of the analgesic effects of flunixin, firocoxib, and meloxicam [14], and one of flunixin, meloxicam, and ketoprofen [15]; another compared the analgesic efficacy of meloxicam oral suspension with a control group [16], and the last one compared the analgesic effects of butorphanol tartrate and phenylbutazone administered alone and in combination [17].

The other six studies referred to colic-related abdominal pain. Of the total colic studies, four were experimental, while the other two were clinical trials. Three experimental studies referred to the recovery of mucosal barrier and the analgesia effect in the model of ischemic-injured jejunum. One study compared the effect of firocoxib and flunixin [20], while another compared the effect of flunixin and etodolac [23], and a third investigated the effect of meloxicam and flunixin meglumine [21]. The last experimental study investigated the analgesia efficacy of meloxicam in a model of low-dose-endotoxin-induced pain with lipopolysaccharide (LPS) [22].

Both randomised clinical trials (RTC) investigated horses affected by strangulating small intestinal lesions; one compared flunixin meglumine with meloxicam [18], and the other flunixin meglumine with firocoxib [19].

### 3.2. Characteristics of the Excluded Studies

A total of eight studies were excluded after full-text evaluation. One was a narrative review [24]; the other was an experimental study in ponies [25]. All other studies were excluded due to the lack in pain score, that represents the main outcome of this systematic review.

### 3.3. Analgesic Effects of NSAIDS on the Treatment of Castration-Related Abdominal Pain

The characteristics and study design features of the castration studies are reported in detail in Table 3.

The study of Gobbi et al., 2020, reported increased stiffness and scrotal swelling scores for two horses in both the meloxicam and the firocoxib group from day 1 to day 3. The flunixin group showed statistically significant lower HR compared with the other two groups, while no difference was reported for the RR.

Lemonnier et al., 2022, used a modified post-abdominal surgery pain assessment scale (PASPAS) [26], and showed that there was no significant effect of the NSAIDs on overall pain scores; however, a higher pain score was reached in the second pain score, 3.1 h after ketoprofen compared to flunixin and meloxicam. Physiological parameters were not included in the statistical analysis.

Olson et al., 2015, showed that the median behaviour and visual analogue scores (VAS), as well as stiffness and scrotal swelling scores, were significantly greater in control animals compared to meloxicam-treated animals at all time periods. Physiological parameters were not included in this study.

Sanz et al., 2009, reported no significant difference for the numerical rating scale (NRS) and VAS data among groups. However, the VAS scores were different over time in the three groups. The highest VAS scores were evident at 4 and 8 h after surgery. HR and RR did not show significant difference over time or between the three groups.

### 3.4. Analgesic Effects of NSAIDS on the Treatment of Colic-Related Abdominal Pain

Of the colic-related abdominal pain studies, four were experimental and two were clinical trials. The characteristics and study design features of colic studies are reported in detail in Table 3.

**Table 3 animals-13-03447-t003:** Characteristics and study design features.

Reference	Intervention Dose/Route of Administration/Interval	Intervention Duration/ Additional Interventions	Pain Scores in Different Treatment Groups	Interval Time Pain Score/Assessor Blindness	Behavioural Score	Physiological Parameters
Gobbi et al., 2020 [14]	FL group: 1.1 mg/kg, IV, SID FX group: 0.1 mg/kg, IV, SID MX group: 0.6 mg/kg, IV, SID	7 days/none	FX group = MX group < FL group	Every 24 h/Not specified	Not performed	HR: FX group = MX group> FL group RR: FX group = MX group = FL group
Lemonnier et al., 2022 [15]	FL group: 1.1 mg/kg IV, SID MX group: 0.6 mg/kg, IV, SID KT group: 2.2 mg/kg, IV SID	2 days/12.5% of horses in the ketoprofen group received additional analgesia	FL group = MX group < KT group	Every 24 h/Yes	Included in the pain score	Not included in the statistical analysis.
Olson et al., 2015 [16]	NaCl group: 2 mL/50 kg, PO, BID MX group: 0.6 mg/kg, PO, SID	4 days/none	MX group < NaCl group	Every 24 h/Yes	MX group < NaCl group	Not performed in this study.
Sanz et al., 2009 [17]	BT group: 0.05 mg/kg IM, q 4 h for 24 h after surgery PB group: 4.4 mg/kg IV before surgery, 2.2 mg/kg PO q12 h for 3 days BT-PB group: dosages as above	3 days/none	BT group = PB group = BT − PB group	On days 1, 3, and 4, q6 h; on day 2, q4 h/Yes	Included in the pain score	HR: BT group = PB group = BT − PB group RR: BT group = PB group = BT − PB group
Naylor et al., 2014 [18]	FL group: 1.1 mg/kg IV, BID MX group: 0.6 mg/kg, IV, BID	4 days/32% of horses in the MX group needed additional analgesia, compared to 16% of horses in the FL group	FL group < MX group	Every 12 h/Yes	Included in the pain score	HR: FL group = MX group RR: FL group = MX group Data reported only at admission
Ziegler et al., 2019 [19]	FL group: 1.1 mg/kg, IV, BID FX group: 0.3 mg/kg, IV, loading dose, 0.1 mg/kg, IV SID	6 days/9% of horses in FX group, needed additional analgesia, compared to 27% of horses in the FL group	FL group < FX group	Every 12 h for the first 3 days, then every 24 h for the following 3 days/Yes	Included in the pain score	HR: FL group = FX group RR: Not reported
Cook et al., 2009 [20]	NaCl group: 1 mL/50 kg, IV FL group: 1.1 mg/kg, IV, BID FX group: 0.09 mg/kg, IV, SID	1 day/None	FX group = FL group < NaCl group	4 and 8 h after the end of 2 h of jejunal ischemia/Not specified	Included in the pain score	Not performed in this study.
Little et al., 2007 [21]	NaCl group: 1 mL/50 kg, IV FL group: 1.1 mg/kg, IV, BID MX group: 0.6 mg/kg, IV, SID	1 day/none	MX group = FL group < NaCl group	2, 8, and 16 h after the end of 2 h of jejunal ischemia/Not specified	Included in the pain score	HR: FL group and MX group < NaCl group RR: MX group < FL group < NaCl group at 16 h after surgery
Urayama et al., 2019 [22]	NaCl group: 2 mL/50 kg, PO MX group: 0.6 mg/kg, PO, SID	1 day/100% of horses received FL 1.1 mg/kg at the end of the experiment.	MX group < NaCl group	Every 30 min after LPS injection for the first 120 min, then every 60 min for the following 240 min/Not specified	Included in the pain score	HR: MX group = NaCl group RR: MX group = NaCl group
Tomlinson et al., 2004 [23]	NaCl group: 12 mL, IV, BID FL group: 1.1 mg/kg IV, BID ET group: 23 mg/kg IV, BID	1 day/100% of horses had butorphanol for the first 8 h after pain score (0.05 mg/kg, IM, q 4 h)	ET group = FL group < NaCl group	2 and 18 h after the end of 2 h of jejunal ischemia/Not specified	Included in the pain score	Not performed in this study.

FL, flunixin; FX, firocoxib; MX, meloxicam; ET, etodolac; BT, butorphanol; PB, phenylbutazone; KT, ketoprofen; IV, intravenous; IM, intramuscularly; PO, per os; SID, once daily; BID, twice daily; q, every; h, hours; min, minutes; HR, heart rate; RR, respiratory rate.

### 3.5. Experimental Studies

In the study by Cook at al., 2009, the behavioural pain scoring system [27] showed significantly higher scores at 4–8 h after surgery in horses in the saline group, compared with horses in the flunixin or firocoxib group. The pain scores of flunixin and firocoxib were not significantly different at any time point. The pain scores at 16 h after surgery were not significantly different between groups or from the scores before surgery. Physiological parameters were not included in this study.

Tomlinson et al., 2004, showed that the median behavioural pain score [27] at 2 h that was greater in the saline group compared with the others, and no difference was present between the etodolac and the flunixin group. At 18 h, the pain scores had decreased in all groups, with the saline group still showing a greater score compared with the others, and no difference was present between the etodolac and the flunixin group. Physiological parameters were not included in this study.

Little et al., 2007, reported no significant difference in the total behavioural pain scores [27] between the flunixin and the meloxicam group. The heart rate was not compared between the flunixin and the meloxicam group. In horses treated with flunixin, the heart rate was significantly increased at 8 h compared with preoperative values, while the heart rate was not significantly increased in horses treated with meloxicam at any time point after surgery. The respiratory rate at 16 h after surgery was significantly lower in horses treated with meloxicam, compared with horses treated with flunixin.

Urayama et al., 2018, showed that in the meloxicam group, the pain scores began to rise after 60 min and then remained constant. In the meloxicam group, the behavioural score [27] was significantly lower compared with the saline group at 60, 90, 120, and 180 min. No significant differences in heart rate or respiratory rate were recorded between the two groups (the data were not shown).

### 3.6. Clinical Trials

No significant difference was detected by Ziegler et al., 2019, in behavioural pain scores [27] between the two groups, and there was no significant difference in the use of additional pain control. However, in the firocoxib group, 9% of horses received an additional pain killer, compared with 27% in the flunixin group. Although the relative risk increased threefold, the result was not statistically significant. No significant difference in the heart rate was found between the two groups. No data were reported for the respiratory rate.

Naylor et al., 2014, reported that 16% (5/32) of horses receiving flunixin and 32% (9/28) of those receiving meloxicam were administered additional analgesia. There was no effect of the treatment on the behavioural or social pain score [27]. There was an effect of the centre on pain score, with fewer horses at one centre showing signs of gross pain and having significantly lower postural pain scores. When the pain score was broken down into composite parts, the horses of the flunixin group experienced significantly less pain compared to the horses in the meloxicam group. There was no difference in heart and respiratory rates at admission to the hospital, but, unfortunately, no data were reported for the post-operative period.

## 4. Discussion

The present review of the current literature stems from the consideration of why, despite the availability of more recent NSAIDs, with supposedly fewer side effects, flunixin is still the most widely used NSAID in the UK, USA, and South Africa [7,8]. Surprisingly, the review did not find scientific evidence supporting this, and research investigating the duration and efficacy of NSAIDs in horses is still sparse [28]. In this systematic review, a total of 184 horses were investigated in castration studies and a total of 175 horses suffering from colic. Despite the widespread use of NSAIDs worldwide and the lengthy research period examined (1989–2023), the total number of horses investigated seems relatively small. Looking at the study design of the selected studies, only 3 out of 10 reported a power analysis for the calculation of the sample size. This reduces the probability that the conclusion of a study reflects the true effect [29]. A practical example is given by the clinical trials of Naylor et al. (2014) and Ziegler et al. (2019) with a sample size of 60 and 56 horses, respectively, where the authors highlighted that 164 and 500 horses would have been required to show with 80% power the relevant difference between the groups investigated. Indeed, a small sample size coming from single studies warrants a cautious interpretation of the results. Also, the type of study, whether experimental or not, prospective or retrospective, or randomised, and the randomisation method should influence result interpretation. In this systematic review, two out of six studies on colic pain and all four castration pain studies were clinical trials. Of the castration studies, Lemonnier et al. (2022), Olson et al. (2015), and Sanz et al. (2009) were prospective randomised, controlled, blinded studies showing a good level of evidence (II-LoE), while Gobbi et al. (2020), because of unclear blindness, was considered III-LoE according to the Oxford Centre for Evidence-based Medicine [12]. An absence in randomisation or a lack in the reliability of the randomisation method, as the use of a flipped coin reported in Ziegler et al., 2019 (III-LoE), can lead to an overestimation as well as an underestimation of treatment effects [30]. Moreover, attention must also be paid to the question whether experimental studies can reflect clinical practice. Indeed, castration as an elective non-corrective intervention shows very good correlation with clinical practice while experimental models of colic pain are less reliable in mimicking naturally occurring colic pain. In four out of six studies, the researchers used models of induced disease with vascular ligation of the small bowel (strangulation model) and injection of LPS for the septic colic model. Given the experimental nature of these studies, they probably just represent a vague approximation of natural disease, and therefore the results should be interpreted with extreme caution [31].

Concerning the intervention, a total of 80% of the studies selected used flunixin as the NSAID. Indeed, over time, flunixin is still considered as a sort of gold standard, against which the efficacy of other drugs can be compared. In all castration studies, NSAIDs were used IV at licensed doses and ranges. Only Olson et al., 2015, administered meloxicam orally through an oral suspension that provided, after 1 h from administration and for 24 h, a plasma concentration that exceeded the established 50% of maximum response (EC50) of 0.20 μg/mL [32,33]. Gobbi et al., 2020, adopted a dose of firocoxib of 0.1 mg/kg once a day without the advised loading dose of 0.3 mg/kg on the first day [9], obtaining good results regarding post-operative analgesia. Sanz et al., 2009, observed no additional analgesic effect in the group receiving phenylbutazone compared to the group without phenylbutazone. This is probably due to the more pronounced analgesic effect of phenylbutazone for orthopaedic conditions [7,34].

In the colic pain studies, flunixin was used at double the licensed dose. In the study of Naylor et al. (2014), both drugs, flunixin and meloxicam, were administered off-label at double the dose indicated by the European Agency for the Evaluation of Medicinal Products. Ziegler et al., 2019, used a double dose of flunixin compared with firocoxib administered at 0.1 mg/kg IV after a non-licensed loading dose of 0.3 mg/kg. However, a pharmacokinetic reason supports this choice, as firocoxib does not reach steady state concentrations within the first 72 h without a loading dose [35]. Also, in the experimental studies of Cook et al. (2009), Little et al. (2007), and Tomlinson et al. (2004), flunixin was used at a off-label dose of 1.1 mg/kg IV twice daily. Cook et al., 2008, found no difference in the post-operative pain score of firocoxib compared to flunixin, despite the fact that firocoxib was administered at a lower dosage (0.09 mg/kg IV once daily) and without any loading dose, compared to the clinical trial of Ziegler et al., 2019. This raises the concern about how reliable the indicated licensed doses are for routine NSAID administration. Tomlinson et al., 2004, compared flunixin with etodolac 23 mg/kg IV twice a day, which is relatively COX-2-selective in horses with more sustained efficacy for orthopaedic conditions [34]. However, to the authors’ knowledge, the administration of etodolac has fallen into disuse over time. Another aspect to consider is the effect of additional analgesia, as shown in the prospective clinical trials of Naylor et al. (2014) and Ziegler et al. (2019), where an additional dose of flunixin before surgery could have influenced the pain score.

Regarding pain assessment, different scales, methods, and intervals were used in the selected studies. Behavioural scales, i.e., NRS and VAS, were the most represented pain scales. On the other hand, Olson et al., 2015, used a scale adapted to control post-castration pain [36], while in Lemonnier et al., 2022, horses were evaluated using an adapted post-abdominal surgery pain assessment scale (PASPAS) [26]. From a methodological point of view, the reproducibility, reliability, and validity of the used pain scale, as well as the choice of the observer, can influence the soundness of the results. The choice of the observers affects the reproducibility that is strongly correlated with intra- and inter-observer reliability [37]. Higher values of intra- and inter- observer reliability indicate a higher precision of the measurements taken by each observer [38]. In fact, the PASPAS is reported to be a reliable tool with low inter-observer variability when expert observers are involved [26]; however, as shown by Lemonnier et al., 2022, inter-observer variability drastically increases when non-expert observers are involved. A high inter-observer variability negatively influences the inter-observer reliability and ultimately the reproducibility of the test. Only 30% of the studies reviewed reported the number of observers and their degree of experience, impairing the reliability of the pain score. Furthermore, only two out of four studies on castration and two out of six on colic pain reported blindness to the intervention of pain score assessors, resulting in a remarkable increase in risk of bias. Another important fact is the time interval of pain scoring that was very variable between studies, with some studies giving a pain score only once a day. Validity represents the ability of a pain score to measure what it is supposed to measure, with minimal inter- and intra-observer variability [37]. To the best of the authors’ knowledge, only Lemonnier et al. (2022) adopted a properly validated pain score. Urayama et al. (2019) described the behavioural pain score [27] used in their study as validated, but to the authors’ knowledge, this pain score has not been validated. Physiological parameters, such as heart rate and respiratory rate, were also analysed in some studies, even though they are non-specific for the presence of pain, and studies have failed to establish a direct relation between heart rate and the presence or severity of pain [39]. Factors such as ambient temperature, dehydration, excitement, and cardiovascular and/or respiratory disease can trigger a physiological response and increase bias [40]. Bias in clinical trials can be defined as a systematic error that can promote one outcome over another and lead investigators to the wrong conclusions about the effects of selected interventions [41]. In the included studies of this systematic review, selection, performance, and “other” bias were the most frequently encountered types of bias. The first one was due to the absence of a clear randomisation method in 70% of the studies, and to the lack in allocation concealment, that is an important step for an adequate randomisation [41]. The detection bias was because the assessors were not blinded to the outcome in 50% of the studies. Moreover, none of the studies specified whether caregivers were also blinded to the selected intervention, generating performance bias. Finally, the “other bias”, that represents the main bias for selected outcomes of this systematic review, was the absence of a validated pain score.

Several limitations are present in this systematic review, such as the choice of the SYRCLE RoB tool [13], the inclusion of experimental studies, and also the string for the search strategy that might have led to the loss of some relevant studies. The modified SYRCLE’s risk of bias tool was selected for continuity with the Systematic Review Protocol for Animal Intervention Studies. The SYRCLE RoB tool has been developed for experimental animal studies and is therefore not ideal for clinical trials. As other tools are based on human RCTs, their application to an animal setting could itself be a source of bias. Because experimental animal studies were included in this systematic review, the choice of the SYRCLE RoB tool was considered appropriate. As suggested by the SYRCLE RoB tool, criterion questions were adapted for bias research to suit the needs of this systematic review. However, no mention of how the modification may result in the development of bias is present. The criterion questions number 4 and 9 were readjusted for the needs of this systematic review, and question number 6 was judged not to be applicable because it was highly linked to laboratory animal studies. However, it is the authors’ impression that this represents a possible source of bias that could undermine the level of evidence at which a systematic review is aimed. Therefore, it would be desirable that specific tools for bias will soon be available for the evaluation of veterinary RCTs. In theory, clinical interventions should only be used if they have been proven safe and effective in well-structured studies. However, this systematic review shows how evidence-based decisions often result from underpowered randomised studies and with unclear control of bias. Still, it is the clinicians who must decide whether they believe that the intervention should be used or not in clinical practice. The latter represents an interesting point, as it appears that over the years the focus of NSAID scientific evaluation has changed its direction. In fact, in the selected studies especially for colic pain, considerable attention was paid on the anti-inflammatory and pharmacological effects on the enteric mucosa rather than on analgesic efficacy. However, clinically, the use of NSAIDs is primarily still aimed at achieving an expected analgesic effect rather than selecting the best NSAIDs with regard of COX selectivity.

## 5. Conclusions

Experimental studies have clearly shown that concerning mucosal interference, COX-non-selective NSAIDs are worse than COX-selective ones; however, COX-non-selective NSAIDs are still the most frequently used drugs in a clinical setting. Therefore, the present study aimed at answering the question: “What is the clinical efficacy of NSAIDs in terms of analgesia?”. The answer is that to date, the available studies cannot adequately address this question, as for many of them, the pain score was not the main outcome but a secondary component. Therefore, new prospective randomised blinded clinical trials, focusing on addressing pain, with a validated easy-to-use pain score, are deemed necessary to elucidate the analgesic efficacy of NSAIDs in the treatment of abdominal pain in horses.

## Figures and Tables

**Figure 1 animals-13-03447-f001:**
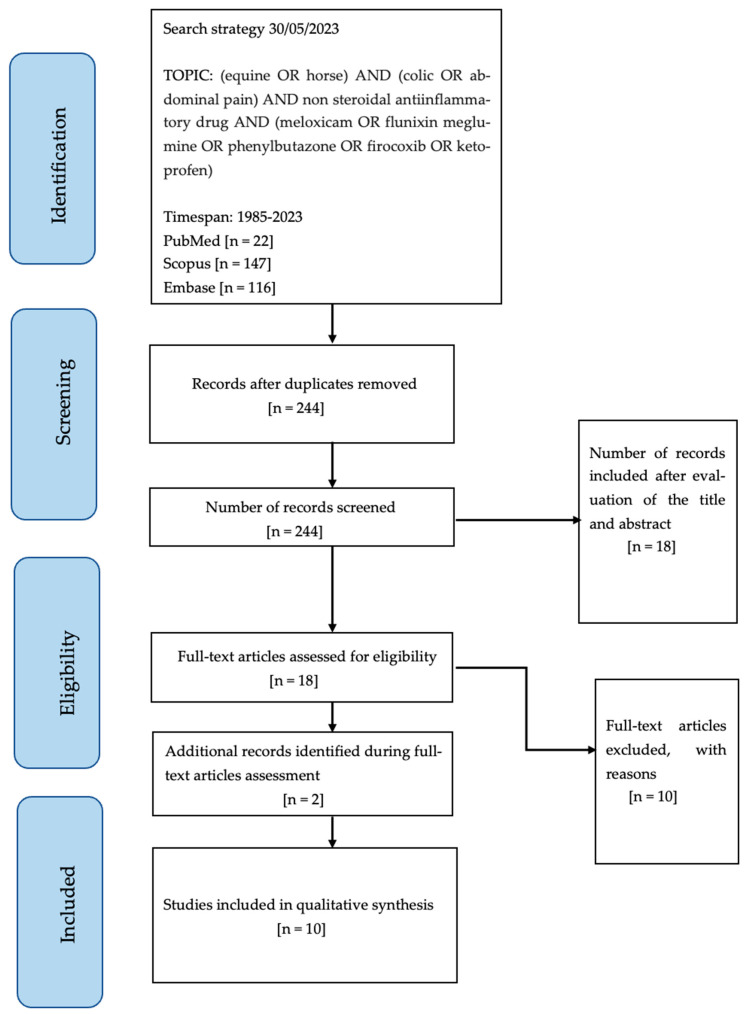
Study flow diagram.

**Table 1 animals-13-03447-t001:** Assessment of risk of bias and level of evidence.

Criterion Question	Gobbi et al., 2020 [14]	Lemonnier et al., 2022 [15]	Olson et al., 2015 [16]	Sanz et al., 2009 [17]	Naylor et al., 2014 [18]	Ziegler et al., 2019 [19]	Cook et al., 2009 [20]	Little et al., 2007 [21]	Urayama et al., 2019 [22]	Tomlinson et al., 2004 [23]
Was the allocation sequence randomly generated and applied?	Yes	Yes	Yes	Yes	Yes	Yes	Unclear	Unclear	Yes	No
Was the distribution of relevant baseline characteristic balanced for the intervention and control groups?	Yes	Yes	Yes	No	Yes	Yes	Yes	Yes	Yes	Yes
Was the allocation adequately concealed?	Unclear	Unclear	Unclear	Unclear	Unclear	No	Unclear	Unclear	Unclear	Unclear
Were the pain scales validated?	No	Yes	No	No	No	No	No	No	No	No
Were the caregivers and/or investigators blinded from knowledge which intervention each animal received during the experiment?	No	No	Yes	Yes	No	Yes	Unclear	Unclear	Unclear	Unclear
Were animals selected under randomization	Not applicable	Not applicable	Not applicable	Not applicable	Not applicable	Not applicable	Not applicable	Not applicable	Not applicable	Not applicable
Was the outcome assessor blinded?	Unclear	Yes	Yes	Yes	Yes	Yes	Unclear	Unclear	Unclear	Unclear
Were incomplete outcome data adequately addressed?	No	Yes	No	No	Yes	Yes	Yes	Yes	No	Yes
Is the outcome of the study appropriate to answer the present review’s question?	No	Yes	No	No	No	No	No	No	No	No
Was the study apparently free of other problems that could result in high risk of bias?	No	No	No	No	No	No	No	No	No	No
Level of evidence (LoE)	IIILoE	II-LoE	II-LoE	II-LoE	II-LoE	IIILoE	IIILoE	IIILoE	IIILoE	IIILoE

**Table 2 animals-13-03447-t002:** Study features of standardised methodological assessment of the included studies.

Reference	Gobbi et al., 2020 [14]	Lemonnier et al., 2022 [15]	Olson et al., 2015 [16]	Sanz et al., 2009 [17]	Naylor et al., 2014 [18]	Ziegler et al., 2019 [19]	Cook et al., 2009 [20]	Little et al., 2007 [21]	Urayama et al., 2019 [22]	Tomlinson et al., 2004 [23]
Journal	*Journal of Equine Veterinary Science*	*Animals*	*Journal of Equine Veterinary Science*	*Journal of the American Veterinary Medical Association*	*Equine Veterinary Journal*	*Equine Veterinary Journal*	*American Journal of Veterinary Research*	*American Journal of Veterinary Research*	*Journal of Equine Veterinary Science*	*American Journal of Veterinary Research*
Study design	Prospective randomized clinical trial	Prospective randomized, blinded clinical trial	Prospective randomized, blinded, controlled study	Prospective randomized, blinded, clinical trial	Prospective randomized, blinded, clinical trial	Prospective randomized, blinded, multicentre clinical trial	Experimental randomized, study	Experimental study	Experimental randomized, cross-over study	Experimental study
Total number of horses	Thirty	Thirty	Eighty-eight	Thirty-six	Sixty	Fifty-six	Eighteen	Eighteen	Five	Eighteen
Source of pain	Castration in standing sedation	Castration in general anaesthesia	Castration in standing sedation	Castration in general anaesthesia	Colic in general anaesthesia	Colic in general anaesthesia	Colic in general anaesthesia	Colic in general anaesthesia	Colic due to lipopolysaccharide injection without sedation	Colic in general anaesthesia
Group number/control group	Three/None	Three/None	Two/Yes	Three/None	Two/None	Two/None	Three/Y es	Three/Y es	Two/Yes	Three/Yes
Intervention	Flunixin, firocoxib, and meloxicam	Flunixin, ketoprofen, and meloxicam	Meloxicam	Phenylbutazone	Flunixin, meloxicam	Flunixin, firocoxib	Flunixin, firocoxib	Flunixin, meloxicam	Meloxicam	Flunixin, etodolac
Power analysis	No	No	Yes	No	Yes	Yes	No	No	No	No
Pain score/validation	Stiffness and scrotal swelling score/Not validated	Modified post-abdominal surgery pain assessment scale/Validate and adapted	Behavioural pain score, visual analogue score, accelerometers (movement evaluation), stiffnessm and scrotal swelling score/Not validated	Visual analogue scores and numerical rating scale/Not validated	Behavioural pain score/Not validated	Behavioural pain score/Not validated	Behavioural pain score/Not validated	Behavioural pain score/Not validated	Behavioural pain score/Not validated	Behavioural pain score/Not validated
Randomization/method	Yes/Statistical Analysis System, version 9.4	Yes/Randomized matrix	Yes/Not specified method	Yes/Not specified method	Yes/Not specified method	Yes/Flip of a coin	Yes/Not specified method	Not specified	Not specified	Not specified
Behavioural parameters	No	Postural, interactive, and colic behaviour score	Postural and interactive behaviour score	Social and undisturbed behaviour score	Social and postural score	Social and postural score	Social and postural score	Social and postural score	Postural and interactive behaviour score	Behavioural score
Statistical Analysis	Student–Newman–Keuls test	post hoc Tukey tests	Mann–Whitney test	Kruskal–Wallis ANOVA	*t*-tests or Kruskal–Wallis tests	two-way ANOVA	one-way ANOVA and Tukey test	one-way ANOVA	Bonferroni’s post hoc test or a Steel–Dwass test	one-way ANOVA

## Data Availability

Not applicable.

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
