# Peer review of "Analgesic Efficacy of Non-Steroidal Anti-Inflammatory Drug Therapy in Horses with Abdominal Pain: A Systematic Review"

_animals, 2023, doi:10.3390/ani13223447_

Round 1

Reviewer 1 Report

Comments and Suggestions for Authors

The topic is clearly interesting for equine clinician. This systematic review investigates the analgesic efficacy of different NSAIDs in horses with experimentally induced or naturally occurring abdominal pain, which is currently debated by many equine clinicians.

The study is clear, studies included are well presented and discussed. Although no straight conclusion can be taken in favour of one or another NSAIDs, they are still interesting aiming for more prospective randomised blinded clinical trials in this direction.

Just a few orthographic mistakes detected:

line 54: repetition of "involve"

Figure 1: In the text (lines 168-171) it is stated that "the first selection considering title and abstract included 18 papers. Of these 10 were excluded. After full text examination another 2 studies that met the inclusion criteria were found from references cited, not detected via initial string. Finally, a total of 10 studies were included in this systematic review." However in figure 1 it is reported that from the 18 full-text articles assessed for eligibility [18], 8 were excluded, 2 were added with a final results of 10 studies included in qualitative synthesis.

line 237: spelling mistake with "form" instead of "from"

Author Response

The reply to the Reviewer's comments can be found as an attachment. 

Reviewer 2 Report

Comments and Suggestions for Authors

The manuscript "Analgesic efficacy of non-steroidal-anti-inflammatory drug therapy in horses with abdominal pain: A systematic review" covers a relevant subject in equine medicine and analgesic management. The manuscript is straight forward, scientifically sound and concise with a good discussion. Congratulations! However, there are some minor comments, which should be addressed by the authors.

line 11: it has been shown that COX2 is not only induced by inflammatory processes, it also has some physiologic, homoeostatic properties i.e. in the kidney. Should be adapted.

line 54 delete 1 involve

line 87ff here the first time post castration abominal pain occurs, it would be good to shortly mention this type of pain also in the introduction. It is colic in the sense of abdominal pain, but often not included in the classic intestinal colic idea.

line 93 no negative control group or no control group. Delete the "in" after Therefore,...

line 116 replace.".. consisted in..." by consisted of

line 148 scooring?

Table 2: the expression Type of Pain in row 5 is not really correct here, it is more the origin of pain, the causig event or a similar wording.

line 345 licensed doses

line 346ff  I think also in the Naylor study it was not possible to rule out that horses had flunixin before being referred to the clinic.

l 346-354 Single sentences should not be a paragraph.

l 409-419 Maybe you should also disucss, that a validated pain score performing also in the hands of non-specialists in a reliable and easy to use way might be necessary and also a consensus on it, if we want to do proper clnical trials.

EBM also includes the clinical experience with a drug and that is what you here from clinicians that flunixin is more effective than meloxicam or firocoxib

Author Response

(The authors gave the same response as above.)

Reviewer 3 Report

Comments and Suggestions for Authors

thank you for a well-done paper, this study points to the inadaquate information we have on the use of NSAIDs for the treatment of pain in horses, and the need for well-done RTC studies.

Are there any validated pain scores for the horse? Should there be a universal pain scale to provide better analgesic scoring in studies?

LIne 54 remove one of the "involved" words, have two, need only one

Line 387 "not" blinded

Author Response

(The authors gave the same response as above.)

Reviewer 4 Report

Comments and Suggestions for Authors

Thank you for your manuscript. It is very interesting and it is very well written. I have made some minor comments regarding the some English things and a sentence that is not clear. 

Well done.

Comments on the Quality of English Language

See above

Author Response

(The authors gave the same response as above.)
